# WHY DO EMBEDDING SPACES LOOK AS THEY DO?

## ABSTRACT

The power of embedding representations is a curious phenomenon. For embeddings to function effectively as feature representations, there must exist substantial latent structure inherent in the domain to be encoded. Language vocabularies and Wikipedia topics are human-generated structures that reflect how people organize their world, and what they find important. The structure of the embedding spaces should thus reflect the human evolution of language formation and the cultural processes shaping our world.

This paper studies what the observed structure of embeddings can tell us about the natural processes that generate new knowledge or concepts. We demonstrate that word and graph embeddings trained on standard datasets using several popular algorithms consistently share two properties: (1) a decreasing neighbor frequency concentration with rank that parallels the Matthew effect in spatial dimension, and (2) particular clustering velocities and power law-based community structures. We then assess a variety of generative models of embedding spaces according to these criteria, and conclude that incremental insertion processes based on the preferential attachment-type mechanisms dependent on spatial context best model the phenomena observed in language and network data.

## 1 INTRODUCTION

Vector representations (Mikolov et al., 2013a;b; Pennington et al., 2014; Perozzi et al., 2014) of feature spaces (embeddings) have proven amazingly effective at capturing the properties of discrete entities such as vocabulary words and the vertices of networks capturing human interactions, such as citation graphs or Wikipedia pages. Such embeddings are a foundational tool in machine learning, providing a natural data representation easily applied for classification and regression models.

The power of embedding representations is a curious phenomenon. That embeddings function so effectively as feature representations implies there must exist substantial latent structure inherent in each domain to be effectively encoded. Language vocabularies and Wikipedia topics are representative human-generated structures that reflect how people organize their world, and what they find important. The structure of the resulting embedding spaces thus must reflect the human evolution of language formation and the cultural processes shaping our world.

This paper addresses a big picture question: *what does the observed structure of embeddings tell us about the processes by which people generate new knowledge or concepts*? Word and graph embeddings provide reduced-dimensional "geometric" representation of the meaning of words and conceptual entities. The widespread success of predictive models trained on top of embeddings suggest that they accurately capture the underlying semantics of each entity. The question we study in this paper concerns which knowledge-generation models best explain properties observed in real-world embeddings.

The vocabulary of every language evolves through a series of processes: new words get created to identify new concepts or objects, while old words get repurposed with new senses and meanings. But how/where does this creation happen? Word usage frequencies are governed by a power law distribution, namely Zipf's law (Zipf, 1936; 1949). Do new words tend to be closely associated with high frequency words, perhaps adding shades of meaning to popular terms? Do they tend to fill large empty regions of embedding space, or add to dense clusters of points? If embeddings do indeed capture the meaning of words, then studying the shape of embedding spaces should inform on the processes underlying the evolution of language.

A similar set of questions can be asked about the evolution of cultural, historical and scientific knowledge, as reflected by the contents of Wikipedia. Each entity page in Wikipedia links to other entities, creating a network well represented by its graph embedding. Do high-degree historical entities disproportionately associate with other high-degree entities? Do new historical figures emerge in the context of clusters of earlier entities, or do they fill empty niches in the cultural landscape?

A variety of generative models can be proposed to build embedding-like spaces by incrementally inserting new points positioned through a specific algorithmic process. We seek to compare the properties of such generated spaces against those of observed embedding spaces, to gain insight into the processes underlying human knowledge generation activities. This is complicated by several factors: (1) we cannot directly observe or reconstruct the historical processes that created the entities of observed data sets, (2) there is no direct correspondence between the labels of generated and observed points, and (3) the scale and range of real/synthetic embeddings over different models, algorithms, and datasets are not directly comparable. Thus assessments must be done through non-parametric statistics quantifying similarities of unlabeled point sets. Our contributions here include:

- **Generative processes for embedding spaces.** We define and explore the space of several plausible processes for the evolution of knowledge embedding spaces, incorporating into our models the mechanisms of preferential attachment as well as the effects of attractive and/or repulsive forces. Certain processes generate networks to embed using standard graph representation learning methods, while others directly generate geometric point sets. Particularly novel is our class of preferential placement models.

- **Non-parametric properties of embedding spaces.** Meaningful comparisons of the properties of observed and generated embedding spaces requires statistics which are robust to transformations as well as being independent of node labels. We have identified two novel and distinct properties of interest which can be rigorously measured on embedding spaces:

  - *Frequency concentration* – How the importance of entities in the local neighborhood of $x$ is related to the importance of $x$ itself? This statistic captures the non-uniform spatial distribution of frequencies observed in an embedding.
  - *Clustering velocity* – How rapidly do clustering procedures advance towards full connectivity? This statistic captures the predominance of cluster structure inherent in an embedding.

  As a check, we also compare these statistics to measures of spatial correlation popular in geographic analysis. Our tools allow to uncover novel regularities in the natural embedding spaces: elements of embeddings are not only characterized by power-law distributions of their frequencies/degrees, but they also exhibit positive assortative sorting in a spatial dimension and a power law manifestation in relational space. Moreover, we show that these observed properties of English language vocabularies and Wikipedia network embeddings consistently hold for a variety of distinct data sources and embedding algorithms, including fastText (Bojanowski et al., 2017), GloVe (Pennington et al., 2014), word2vec (Mikolov et al., 2013a), and BERT (Devlin et al., 2018). Thus they are not mere algorithmic artifacts of data collection or analysis, but inherent properties of language and culturally-generated networks: phenomena in need of explanation through models of language/knowledge creation.

- **Evaluation of generative processes for word and graph embedding spaces.** We compare embeddings generated by several different processes to real data-driven embeddings from several datasets by these criteria. Several models can easily be eliminated based on these observations, leaving few that remain plausible. We conclude that specific incremental insertion processes based on the spatial context-dependent preferential attachment mechanisms best model the observed phenomenon on language and network data.

**Paper outline.** The rest of this paper is organized as follows. Section 2 defines and presents the collection of seven different models we consider as potential generative processes. Section 3 proposes our frequency concentration statistic and evaluates embeddings from both models and real-world ones. We present our clustering velocity statistics and model evaluations in Section 4. Finally, we discuss the implications of this research and future work in Section 5. Our code and datasets are accessible at https://shorturl.at/dyST6.

## 1.1 RELATED WORK

**Explanations of embedding spaces.** Levy & Goldberg (2014) find that word embedding model such as word2vec essentially factorizes a shifted pointwise mutual information matrix. To further understand the linear relationship of embeddings, Allen & Hospedales (2019); Allen et al. (2019) mathematically define the meaning of statement "*word vector $\boldsymbol{w}_x$ is to word vector $\boldsymbol{w}_y$*" and provide a proof of the linearity among embeddings of word analogies. This linear property has also been explained in knowledge graph embeddings (Allen et al., 2021). However, this line of work is more focusing on the linear property between embedding vectors instead of the processes of generation.

**Generative models of embeddings.** There are many graph/network generative models (see the survey Hamilton (2020) and references therein). Traditional processes include the Erdős–Rényi model (Erdos et al., 1960), stochastic block model (Holland et al., 1983), and preferential attachment models (Simon (1955), Barabási & Albert (1999), also see Albert & Barabási (2002) for a review). Arora et al. (2016) study a generative model where each word vector captures its correlations with the discourse vector. Instead, our generative models employ simple heuristics to generate embedding vectors that exhibit empirically realistic properties. There are also works on generating models by using deep graph neural networks (You et al., 2018). Xiao et al. (2016) propose a model based on Dirichlet Process to generate embeddings of knowledge graph. Li et al. (2016) propose a graphical model for generating embeddings of documents. Most of these models are heavily based on a probabilistic framework.

**Notation.** For a set of $n$ entities (e.g. words or nodes) $\mathcal{V} := \{1, 2, \ldots, n\}$, the set of embeddings associated with $\mathcal{V}$ is $\{\boldsymbol{w}_1, \boldsymbol{w}_2, \ldots, \boldsymbol{w}_n\}$, where each *location* or *whereabouts* $\boldsymbol{w}_i \in \mathbb{R}^d$. Throughout this paper, each entity $i$ is also associated with a *importance* or *prominence* (e.g. the frequency of a specific words, or the degree of a graph node) and denoted as $p_i$. Therefore, each entity $i$ can be represented as pair of importance and location $\langle p_i, \boldsymbol{w}_i \rangle$. We collect them in an embedding matrix $\boldsymbol{W} \in \mathbb{R}^{n \times d}$ and an importance vector $\boldsymbol{p} \in \mathbb{R}^n$.

## 2 GENERATIVE MODELS FOR EMBEDDING SPACES

A generative model of an $n$-element embedding space starts with an initial entity embedding pair $\langle p_0, \boldsymbol{w}_0 \rangle$. At each time $t = 1, 2, \ldots, n$, the model then repeatedly generates a new pair $\langle p_t, \boldsymbol{w}_t \rangle$. During this generating process, it updates $\boldsymbol{p}$ and $\boldsymbol{W}$. We first present simple baseline models to generate points before considering network-generation processes based on preferential attachment. Motivated by empirical facts presented later, we seek to generate an embedding space such that entities with similar properties will be closer with each other compared with more distant points.[1]

**A) Generating nodes of a network**

**A1) Gaussian model.** A simple way to generate an embedding-like space creates each point $\langle p_i, \boldsymbol{w}_i \rangle$ where $\boldsymbol{w}_i$ is from a standard Gaussian distribution $\mathcal{N}(\boldsymbol{0}, \boldsymbol{I}_d)$ and the prominence $\boldsymbol{p}$ is drawn uniformly from a random permutation of $\{1, 2, \ldots, n\}$ so that each entity $i$ has a random rank $p_i$. The right figure illustrates generated embedding points that are from 3-dimensional Gaussian distribution, i.e., $\mathcal{N}(\boldsymbol{0}, \boldsymbol{I}_3)$. We color each point $i$ by its normalized prominence, i.e., $p_i / \max_{i \in \mathcal{V}} p_i$. As expected, these data points form a single cluster without any finer substructure within it.

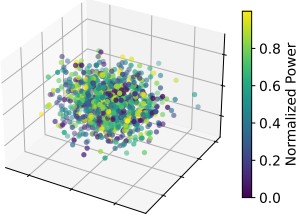

**A2) Preferential Placement (PP) model.** This model presumes that new entities emerge around the older, relatively more prominent ones. This PP model builds on the *preferential attachment* mechanism driving the development of real-world networks and graphs, where the frequency of generated entities exhibit the power law distribution, and it is closest to Simon (1955).[2]

---

[1]We describe experimental details of generating points in Appendix A.1.

[2]The idea can be traced back to Eggenberger & Pólya (1923) and Yule (1925), with accepted modern formulations due to Simon (1955), Price (1976) and Barabási & Albert (1999).

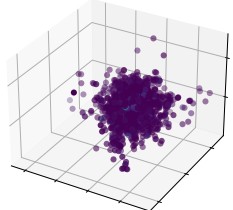

PP is parameterized by the total frequencies of data points $n$, the probability of increasing the frequency of an old entity $q$, and an exponential distribution[3] with scale parameter $\beta$ driving the distance of a new entity from the old one per-iteration $t$. The PP model randomly selects entity $i$ from existing ones with a probability proportional to their frequencies and with a probability $q$, it increases the frequency of $\boldsymbol{w}_i$ by 1. Otherwise, it generates a new entity $\boldsymbol{w}_t$ within a small ball around $\boldsymbol{w}_i$: $\boldsymbol{w}_t = \boldsymbol{w}_i + r_t \cdot \boldsymbol{x}_t$ with $p_t = 1$, where the radius $r_t \sim \exp(\beta)$, and its direction $\boldsymbol{x}_t$ follows a uniform distribution on a unit sphere.[4] Different from the above Gaussian model, the left figure shows generated points where more prominent points in the small core cluster are surrounded by less prominent ones.

**A3) Directional Preferential Placement (DPP) model.** The previous PP model generates new points in uniformly distributed random directions, producing a homogeneous scattering of embedding entities. To bring about a more heterogeneous landscape with cluster structure, we use the *von Mises-Fisher distribution* (vMF) that allows more control over the directions where new entities are generated, in particular making this process more spatial context-aware. Specifically, a random unit vector $\boldsymbol{x} \in \mathbb{R}^d$, $d \geq 2$, follows the $d$-dimensional vMF distribution if its probability density function is given by

$$f_{\mathrm{vMF}}(\boldsymbol{x}|\boldsymbol{\mu}, \kappa) := \frac{\kappa^{d/2-1}}{(2\pi)^{d/2} I_{d/2-1}(\kappa)} \exp\left\{\kappa \boldsymbol{\mu}^\top \boldsymbol{x}\right\},$$

where $\boldsymbol{\mu}$ with $\|\boldsymbol{\mu}\| = 1$, is the mean direction parameter specifying the orientation relatively to the origin; $\kappa \geq 0$ is a concentration parameter controlling the dispersion about the mean (higher $\kappa$ produces higher concentration); and $I_{d/2-1}(\kappa)$ is the modified Bessel function of the first kind. For example, mean direction may represent the force attracting $\boldsymbol{x}$ to some local point, or, say, repelling it from a global center. vMF distribution has been applied for high-dimensional directional data analysis, particularly in applications to textual data and with a focus on clustering (Banerjee et al. (2005) and Gopal & Yang (2014), also see Sra (2018) for a recent overview). However, to the best of our knowledge, prior work considered some given fixed parameters of interest (usually focusing on estimating them, sometimes on using them for generating exercises); while this paper is interested exclusively in data-generating models, and treats at least some of these parameters (e.g., mean direction $\boldsymbol{\mu}$) as endogenously defined within the model itself.

In the DPP model, the first entity $\langle p_0, \boldsymbol{w}_0\rangle$ is initialized with $p_0 = 1$ and $\|\boldsymbol{w}_0\| = 1$. Given the total frequencies of entities $n$, the exponential distribution $\exp(\beta)$ with parameter $\beta$, and the probability of increasing the frequency of an old entity $q$, the DPP model involves (1) selection as an origin of an existing entity $\boldsymbol{w}_i$ randomly with a probability proportional to its frequency; and then (2) at each iteration $t$, with probability $q$, either (2a) an increase of the frequency of the chosen $\boldsymbol{w}_i$ by 1; or (2b) a creation of a new entity around the origin $\boldsymbol{w}_i$, i.e., $\boldsymbol{w}_t = \boldsymbol{w}_i + r_t \cdot \boldsymbol{x}_t$. In contrast to the previous model, however, now the direction is distributed

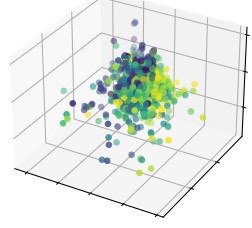

as $\boldsymbol{x}_t \sim \mathrm{vMF}(\boldsymbol{x}|\boldsymbol{\mu}_t, \kappa)$, with mean direction vector $\boldsymbol{\mu}_t$ pointing away from the global center of mass $\boldsymbol{w}_t^*$.[5] The generated points of this model is illustrated on figure to the right where higher weight nodes are close to each other.

**B) Generating edges of a graph.**

**B1) BA-Embedded model.** The BA model uses a preferential attachment process (Barabási & Albert, 1999) to generate networks of edges, as opposed to the previously described preferential placement models for points. We use the BA model to generate a random graph and then apply a standard graph representation learning algorithm to obtain an embedding.

---

[3]Recall that the exponential distribution with scale parameter $\beta$ is denoted as $\exp(\beta)$ and its probability density function is given by $f_{\exp}(x; \beta) = 1/\beta \exp(-x/\beta)$ when $x \geq 0$, 0 otherwise.

[4]That is, $\|\boldsymbol{x}_t\| = 1$. $\|\cdot\|$ is the $\ell_2$-norm, i.e., $\|\boldsymbol{x}\| = \sqrt{\sum_{i=1}^d x_i^2}$. For example, when $d = 2$, the offset term $r_t \cdot \boldsymbol{x}_t = [r_t \cos(\theta_t), r_t \sin(\theta_t)]^\top$, with angle $\theta_t \sim \mathcal{U}(0, 2\pi)$.

[5]We define the center of mass as the weighted average of all points, that is $\frac{1}{t}\sum_{i \in \{1,2,\ldots,t\}} p_i \boldsymbol{w}_i$. This center of mass moves at each iteration $t$.

Denote the number of nodes at time $t$ as $n_t$. Each new node is added to the network and connected to node $n_i$ with probability $q_t = p_i / \sum_j p_j$, where weight $p_j$ is the degree of node $j$.

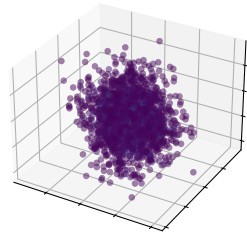

Our BA-Embedded model starts by creating a star graph with $m + 1$ nodes where node $0$ connects all other $m$ nodes, that is, $1/m$ equals to the probability of generating new nodes. Let $\mathcal{L} \leftarrow \{(0,1), (0,2), \dots, (0,m)\}$ be the initial list of $m$ edges. At each iteration $t$, the model generates a new node $i$ and randomly select $m$ targets from current graph, where each edge connects the new node and a node from selected $m$ targets. After creating $m$ new edges, the model adds these edges into $\mathcal{L}$, i.e., $\mathcal{L} \leftarrow \mathcal{L} \cup \{(i, t_1), (i, t_2), \dots, (i, t_m)\}$. We repeat until $n$ nodes are generated and obtain the final edge list $\mathcal{L}$. For generating embeddings, we use DeepWalk (Perozzi et al., 2014) to generate the corresponding embedding vectors with $\mathcal{L}$ as an input graph. As shown in figure left, these generated 3-dimensional points form a cluster where points of smaller weights (less prominent nodes) spread out of the clusters while higher degree nodes are densely correlated. Compared with PP model, lower prominence points of BA-Embedded model are smoother located around the cluster.

**B2) BA model with gravitational movement (BA-Gravity).** Instead of directly using DeepWalk to generate embeddings, we treat the graph generation process as a sequence of edge generation, and then update the embeddings by their weights. Effectively, we extend the BA model with a more sophisticated mechanism of dependence on neighboring spatial context. Algorithm 1 presents the BA-Gravity model. It first obtains a sequence of edges $\mathcal{L}$ of the simulated graph and then initializes each new entity either from $\mathcal{N}(\mathbf{0}, \boldsymbol{I}_d)$ or from $\mathcal{U}[-.5, .5]^d$ (Line 4-9). After list of edges are generated by BA-model, for node $u$, it updates embedding $\boldsymbol{w}_u$

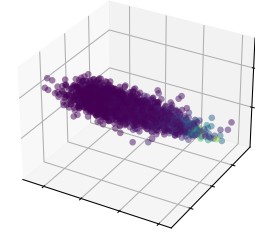

by moving $1/(p_u + 1)$ away of itself but in the direction to $\boldsymbol{w}_v^t$, i.e., $\frac{p_u}{p_u+1}\boldsymbol{w}_u^t + \frac{1}{p_u}\boldsymbol{w}_v^t$. We call this update step "*gravitational movement*" since its neighbor node $v$ attracts $u$ to itself and away from its original location. The generated 3-dimensional points by Algorithm 1 are shown in figure right. Clearly, higher degree nodes are closely located.

---

**Algorithm 1** BA-Gravity model

---

1: $\mathcal{L} \leftarrow$ list of edges by the BA model
2: $V^0 = \{\}, t = 0, \boldsymbol{p} = \mathbf{0}$
3: **for** $e^t(u, v) \in \mathcal{L}$ **do**
4:     **If** $u \notin V^t$ **then**
5:         $\boldsymbol{w}_u^t \sim \mathcal{N}(\mathbf{0}, \boldsymbol{I}_d)$ or $\mathcal{U}(-.5, .5)/d$
6:         $V^{t+1} = V^t \cup u$
7:     **If** $v \notin V^{t-1}$ **then**
8:         $\boldsymbol{w}_v^t \sim \mathcal{N}(\mathbf{0}, \boldsymbol{I}_d)$ or $\mathcal{U}(-.5, .5)/d$
9:         $V^{t+1} = V^t \cup v$
10:    $p_u = p_u + 1, p_v = p_v + 1$
11:    $\boldsymbol{w}_u^{t+1} = \frac{p_u}{p_u+1}\boldsymbol{w}_u^t + \frac{1}{p_u}\boldsymbol{w}_v^t$
12:    $\boldsymbol{w}_v^{t+1} = \frac{p_v}{p_v+1}\boldsymbol{w}_v^t + \frac{1}{p_v}\boldsymbol{w}_u^t$
13:    $t = t + 1$
14: **end for**
15: **Return** $W$

---

Denote $t_v$ as the time of node $v$ being the neighbor of $u$. During the whole process of BA-Gravity, at any time $t$, one can verify that $\boldsymbol{w}_i$ is given by $\boldsymbol{w}_i^{t+1} = \frac{p_u}{p_u+1}\boldsymbol{w}_i^t + \frac{1}{p_u}\boldsymbol{w}_v^t = \frac{1}{p_u+1}\boldsymbol{w}_i^0 + \frac{1}{|\mathcal{N}(i)|}\sum_{j \in \mathcal{N}(i)} \frac{1}{p_u}\boldsymbol{w}_j^{t_j}$. Therefore, one can treat Algorithm 1 as the first-order approximation of $\boldsymbol{U} = (\alpha_0 \boldsymbol{I} + \alpha_1 \boldsymbol{A} + \alpha_2 \boldsymbol{A}^2 + \dots + \alpha_q \boldsymbol{A}^q) \boldsymbol{R}$, where $\boldsymbol{A}$ is the normalized adjacency matrix, $\boldsymbol{I}$ is the identity matrix, and $\boldsymbol{R}$ is a random matrix. This embedding approximation has been explored in fast graph embedding algorithms (Zhang et al., 2018; Chen et al., 2019).

**B3) BA-Centroid(Node)-Average/Weight.** To further extend the BA-Gravity model, instead of updating two nodes, we update the node itself whenever the model creates a new node. That is, at each iteration, it only updates the embedding of the new node by taking average of its neighbors' embeddings. Let $\boldsymbol{w}_i$ be the current embedding to be updated. There are two options of taking this average: 1) simply taking average of all neighbors, i.e., $\boldsymbol{w}_i = \frac{1}{|\mathcal{N}(i)|}\sum_{v \in \mathcal{N}(i)} \boldsymbol{w}_v$ and we refer this model as BA-Centroid-NA; and 2) taking a weighted average, i.e., $\boldsymbol{w}_i = \frac{1}{\sum_{j \in \mathcal{N}(i)} p_j}\sum_{v \in \mathcal{N}(i)} p_v \cdot \boldsymbol{w}_v$. We refer to it as BA-Centroid-NW.

**B4) BA-Centroid(Edge)-Average/Weight.** Compared to BA-Centroid-NA and BA-Centroid-NW, edge-based models update embeddings of both new node and its neighbors by taking average of their neighbors. We call these two methods

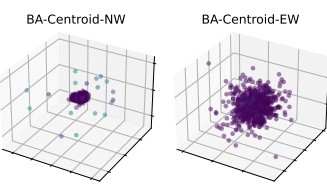

BA-Centroid-NW          BA-Centroid-EW

CENTROID-EA and CENTROID-EW respectively based on the two average strategies above. We illustrate the points of BA-Centroid-NW and BA-Centroid-EW model.

## 3 FREQUENCY CONCENTRATION

New entities emerge in the shadow of older, more prominent entities, creating cluster structures which should be apparent in both model and real-world embeddings such as GloVe (Pennington et al., 2014), word2vec (Mikolov et al., 2013b), and DeepWalk (Perozzi et al., 2014).

Study of relationships between spatially-distributed entities have naturally been an important focus in geography. Similar methods are also relevant for quasi-spatial data in other domains: what is sometimes called the "first law" of geography states that "Everything is related to everything else, but near things are more related than distant things" (Tobler, 1970). Popular measures of spatial autocorrelations include (1) Moran's $I$, which may be thought of as a spatial analog of Pearson's correlation coefficient and is sensitive to extreme values of the variable (Moran, 1948); as well as (2) Geary's $C$, which is sensitive to differences in small neighborhoods (Geary, 1954). A very accessible textbook treatment is available in O'Sullivan & Unwin (2010); also see Anselin (1995). For examples of usage see: in epidemiology Ganegoda et al. (2021), in linguistics Grieve (2011).

We design our own statistical metric capturing the geometric properties of the embeddings introduced earlier in a non-parametric manner.[6] The key idea of the proposed index to measure frequency concentration has three parts. First, we build a $k$-nearest neighbor similarity graph $\mathcal{G}(\mathcal{V}, \mathcal{E})$ based on $d$-dimensional node/word embeddings, i.e., $\boldsymbol{W} \in \mathbb{R}^{n \times d}$. The potential measure function could be either a Gaussian Kernel function or a cosine similarity. The frequencies of the one-hop neighbors of each data point in this graph (e.g., a node or a word) can be interpreted as a frequency rank (from 1 to $n$), and these sum of ranks normalized to get a value from 0 to 1. We define an index from the area under the cumulative distribution function (CDF) of the neighbors' rank as a measure of spatial autocorrelation. More formally, we assume the set of entities considered is $\mathcal{V} = \{1, 2, \ldots, n\}$. Each $v \in \mathcal{V}$ has a prominence associated with it, denoted as $p_v$ (e.g. node degree or word frequency).

**Definition 1** (Frequency concentration index). *Denote the prominence $p_v$ as a vector $\boldsymbol{p} = [p_1, p_2, \ldots, p_n]^\top$ for all nodes in $\mathcal{V}$. Let the ranking function $r(\boldsymbol{p}) : \mathbb{R}^n \to \pi(\mathcal{V})$ where $\pi$ is a permutation of $\mathcal{V}$ where $r_v$ is the ranking of node $v$ taking a value in $[1, n]$. Given $k$ and any embedding matrix $\boldsymbol{W} \in \mathbb{R}^{n \times d}$, we define $\mathcal{N}(v; k, \boldsymbol{W})$ as a set of one-hop $k$-nearest neighbors of $v$ based on some chosen similarity function on $\boldsymbol{W}$. Then the Index of $v$ can be defined as the following:*

$$\text{Index}(v; k, \boldsymbol{W}) := \frac{1}{n} \sum_{i=1}^{n} \left( \frac{|\{r_u \leq i : u \in \mathcal{N}(v; k, \boldsymbol{W})\}|}{k} \right) = 1 - \sum_{i \in \mathcal{N}(v; k, \boldsymbol{W})} r_i. \quad (1)$$

Intuitively, if most of neighbors have high rankings, then the index is close to 1 while if neighbors have low rankings, it is close to 0. Each term in the summation of Equation (1) can be viewed as reflecting the effects of preferential attachment.

**Frequency concentration of word embeddings:** We present the frequency concentration plots for different embeddings including Fasttext (Mikolov et al., 2018), word2vec (Mikolov et al., 2013b), and GloVe (Pennington et al., 2014) in Figure 1, comparing the frequency correlations of English words embedding that are from eight different models and data sources. Figure 1 shows that the frequency concentration is remarkably consistent across different data sources and algorithms, that is: *the higher frequency words tend to have high frequency words as their close neighbors, while low-frequency words live in poorer neighborhoods*. It is tempting to view this as a manifestation of relational, spatial *Matthew effect*.[7] Results in the appendix show similar frequency concentration across a broad range of non-English languages. We also investigate how the frequency correlations change with respect to the parameter $k$ and $d$. More detailed results can be found in Appendix B.1.

**Frequency concentrations of graph embeddings** Table 1 shows that the slopes of the best linear fit are consistently negative across four diverse graph datasets (detailed in Table 3 and Figure 5

---

[6]In the interest of completeness, we also provide results for the classical Moran's $I$ and Geary's $C$ statistics.
[7]Originally christened by Merton (1968), and in our case implying that not only "the rich get richer", but also that "they befriend other riches".

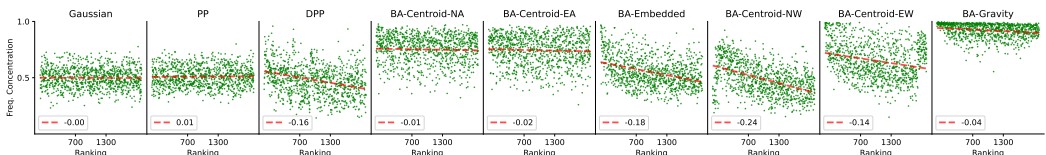

Figure 1: Frequency concentration of eight distinct English word embeddings. The observed frequency concentration is similar even though trained on different data sources (e.g., Wikipedia (Wiki), Common Crawl (CC/42B/840B/Sub), and Google News (News)) and different embedding algorithms including GloVe, fastText, and word2vec. Each dashed red curve is linearly fitted where linear coefficient is presented on each legend.

in the appendix) for four distinct algorithmic approaches to graph embeddings. As with language vocabularies, the frequency concentration plot decreases consistently with rank across diverse embedding models for the all graphs. The consistency of these results across domain and algorithm mark negatively-sloping frequency concentration as a property to be preserved in any meaningful generative model of embeddings.

|            | LLE     | FastRP  | DeepWalk | LINE    |
|------------|---------|---------|----------|---------|
| Coauthor   | -0.1582 | -0.4932 | -0.3780  | -0.2808 |
| Wikipeople | -0.3498 | -0.1730 | -0.2708  | -0.1760 |
| Patent     | -0.5619 | -0.7356 | -0.6705  | -0.5832 |
| PPI        | -0.2739 | -0.5144 | -0.3459  | -0.4092 |

Table 1: The frequency concentration slope of graph embeddings are consistently negative across four algorithms/datasets.

**Model validation.** We evaluate the frequency concentration of proposed generative embedding models in Figure 2, with proximity measured by cosine similarity. Clearly, embeddings of models with fitted negative coefficient are more likely close to real-world embeddings as shown in Figure 1. Only the context-aware variations of preferential placement and BA models match the data. Figure 5 demonstrates that the observed properties of such representations are consistent across four different graph embedding algorithms.

Figure 2: Frequency concentration plots for all nine different models where each model generates $n = 2000$ points with $d = 128$. The parameter $k$ of the Index for each node is set to 10.

## 4 CLUSTERING VELOCITY

The spatial-frequency concentration presented above provides one dimension to compare real-world embeddings to models. We now analyze the spatial characteristics of embeddings for clustering structures independent of frequency, and propose two distinct indices to measure the *cluster velocity* of the $k$-nearest-neighbor ($k$-nn) graph induced from the vector representation space.

Since embeddings $W \in \mathbb{R}^{n \times d}$ are optimized to preserve the natural clusters of the real-world knowledge in its metric space, its pair-wise similarity graph $G_{metric} \in \mathbb{R}^{n \times n}$ encodes the embedding spatial characteristics as vertex connectivity. We hypothesize that the $k$-NN graph $\mathcal{G}$ induced from real-world $G_{metric}$ should share common connectivity patterns, particularly in the number of connected components when we perform agglomerative clustering.

For example, consider a point set drawn from $p$ independent, well-separated Gaussian distributions, and hence containing $p$ natural clusters. We anticipate that any single-link agglomerative clustering procedure from shortest to longest edge will quickly identify these clusters, but many short edges must be evaluated as potential merges before the separate components are unified by long inserted

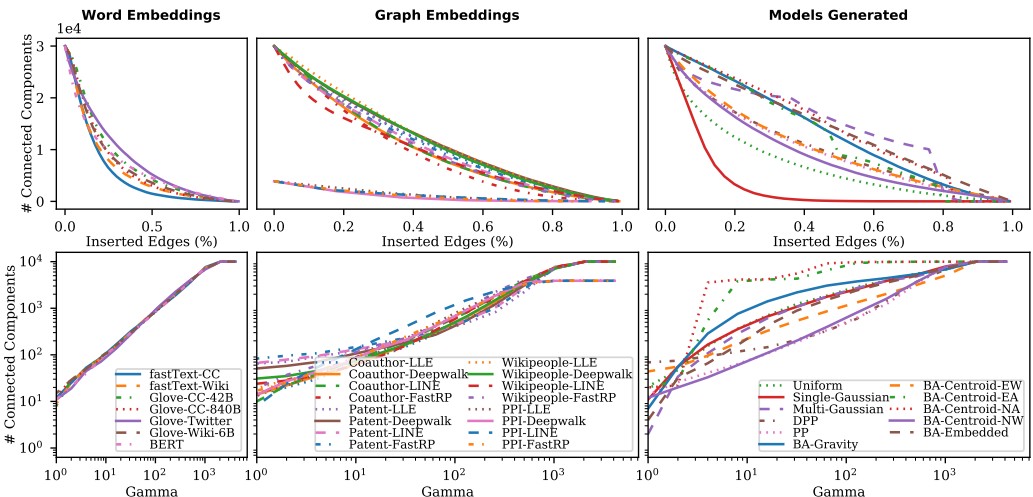

Figure 3: ***Top***: The number of connected components as a function of inserted edges. Each column shares legend. ***Bottom***: The number of connected components as a function of $\gamma$ in graph partitioning methods Clauset et al. (2004) in log-scale. We sampled top 30K and 10K vectors from embeddings to visualize clustering velocities, except for smaller PPI graph which has only 3,980 nodes. All real-world embeddings share similar clustering velocity pattern, while model generated space have distinct shapes.

edges. The number of connected components will decrease at a slow velocity until eventually all vertices are merged into a single integrate component. This merging process will happen more quickly for $p = 1$ than $p = 2$.

**Empirical clustering velocity from real-world and model generated embeddings.** Figure 3 (top) presents the clustering velocity for the word, graph, and model embeddings we have described. As expected the multiple Gaussian model (upper right, curve *multi-Gaussian* in Figure 3 clusters slower than a single Gaussian. While a point set of uniform distribution may not have such hierarchical cluster formation process, thus, yielding distinct characteristics in term of clustering velocity as shown in the same subfigure. Furthermore, we observed all word and graph embeddings (first and second columns of the ***top row***) have similar curve shape, reflecting non-trivial ubiquitous geometric characteristics in the original embedding space.

**Agglomerative clustering velocity index:** We define the following *Clustering Velocity-AUC* index (CV-AUC) to measure the cluster formation characteristic of agglomerative clustering as follows. We construct the K-nearest-neighbor graph using cosine distance as $\mathcal{G} = (\mathcal{V}, \mathcal{E})$. $|\mathcal{V}| = n$, $\mathcal{E} = \{e_{ij}|e_{uv} = dist_{cos}(w_u, w_v); u \neq v; u, v \in \mathcal{V}\}$ [8] Then, given an empty edge set $\tilde{\mathcal{E}} = \emptyset$, we reconstruct the graphs by inserting $|\mathcal{E}|$ edges in ascending order. We denote $f_{\mathcal{G}}(t) = |\{\tilde{S}_1^t, \tilde{S}_2^t, ..., \tilde{S}_l^t\}|$ as the number of connected components, where $\tilde{S}_k^t$ is the $k^{th}$ connected component in graph $\mathcal{G}$ after $t^{th}$ insertion. The Clustering Velocity is defined as the normalized AUC of $f_{\mathcal{G}}(t)$:

$$\text{Index}_{\text{CV-AUC}}(\mathcal{G}) := \int_0^{|\mathcal{E}|} \frac{f_{\mathcal{G}(t)}}{|\mathcal{E}| * |V_{\mathcal{G}}|} \, dt \in (0, 1). \tag{2}$$

**Clauset-Newman-Moore Clustering Velocity (CV-CNM)** : In addition, similar entities tend to have closer proximity in the embedding space, yielding graph clusters having stronger/denser intra-group ties/edges. We are interested in finding how inter/intra-clusters take shape, specifically, how fast the big components branch into sub-clusters as we cut the edges. Therefore, we use a modularity-based graph partitioning algorithm Clauset et al. (2004) as another way to inspect the clustering velocity as we tune resolution parameter $\gamma$ from small to large. As shown in the ***bottom row*** of Figure 3 (in log-scale), likewise, all real-world embeddings share the velocity pattern which is similar to power law, while various models (subfigure on the bottom right) have distinct shapes.

Given $\boldsymbol{W}$ and $\mathcal{G}$ as described in Definition 2, we use modality-based graph partition method to cut $\mathcal{G}$ into sub-graphs $\mathcal{S}^{\gamma} \in \{S_1^{\gamma}, S_2^{\gamma}, ..., S_l^{\gamma}\}$ with varying resolution parameter $\Gamma \in \{\gamma_k | \gamma_k = 2^k, k \in$

---

[8]$dist_{cos}(\cdot, \cdot)$calculates cosine distance

$\mathbb{Z}\}$, where $S_k^\gamma$ is the $k^{th}$ component. We define the index as the error of power-law fitting:

$$\text{CV-CNM} := \frac{1}{|\Gamma|} \sum_{i=1}^{|\Gamma|} (y_k - f'_\mathcal{G}(x_k))^2, \ x_k = \log(\gamma), \ y_k = \log(|\mathcal{S}^\gamma|), \ f'_\mathcal{G}(x) = w_\mathcal{G} * x + b_\mathcal{G},$$

where $w_\mathcal{G}$ and $b_\mathcal{G}$ are the fitted parameters associated with KNN-graph $\mathcal{G}$.

**Model Evaluation:** We visualize these proposed indices, including Frequency concentration, for each embedding in Figure 4 (*left, middle*). On left, the horizontal and vertical axis represents *CV-AUC* and *CV-CNM*, respectively. All the word and graph embeddings of various methods (shown with red and blue markers on *left*) have similar range, demonstrating the existence of *invariant* clustering velocity patterns among real-world embeddings. Although the model generated spaces have distinct values, the *BA-Centroid-{NW,EW}* and *DPP*-family [9] models are demonstrably closer to the real ones.

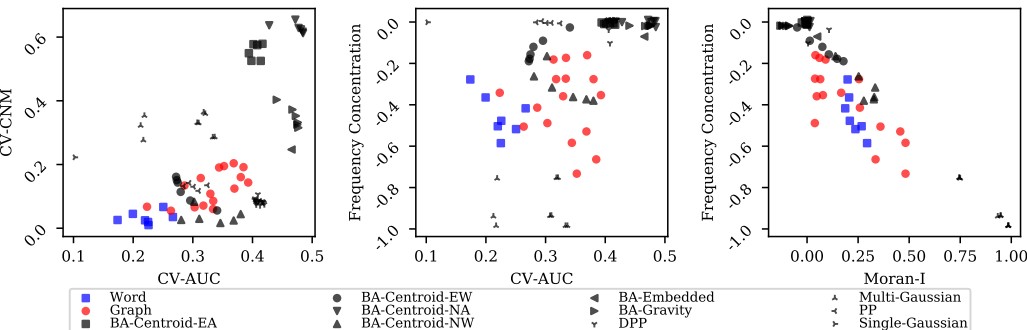

Figure 4: The visualization of real-world and model generated embeddings ($dim = 300$) in the plane of two proposed clustering velocity indices, frequency concentration, and spatial autocorrelation indices . Real-world embeddings share similar pattern as several model generated embeddings (*BA-Centroid-{NW,EW}* and *DPP* ), while other models do not. Each model is represented with five distinct hyper-parameter configurations. On right, frequency concentration correlates well with the spatial autocorrelation statistic Moran's I [-0.810] (and with 1-Geary's C [-0.450] in Figure 9 in Appendix).

# 5 CONCLUSION AND FUTURE WORK

In this paper, we explore invariant spatial characteristics of embeddings and propose several models that reconstruct the observed properties of frequency concentration and clustering velocities of real-world embedding spaces. Our computational tools document a consistent empirical phenomenon (not an artifact) across many word and graph data sets as well as embedding algorithms that requires explanation. Especially interesting findings are the positive association between entities of similar importance, suggesting a spatial kind of Matthew effect, as well as a particular connectivity of the clustering structures that is characterized by a power law in relational sense; which, in turn, seem to be the fundamental processes underlying the formation of clusters in an embedding network.

We present an evolving collection of instructive generating models, starting from simple processes that are agnostic about their local and global neighborhoods and progressing to more complex algorithms that account for spatial interdependence between generated entities. The latter behave like what is observed in typical language/graph datasets, and, as we have shown, position points naturally in correlated patterns and into cluster structures akin to those observed in the wild.

Our work raises several interesting questions. Our experiments were not of sufficient resolution to identify a single dominant model for reasoning about the origins and structure of natural embedding spaces – but they are a start. We believe a well-supported natural model will be a substantial contribution to understanding the evolution of knowledge structures in several domains.

An interesting direction of research would be to study the effects of spatial dependencies within the generating process, the contribution of different attractive and repulsive forces, the importance of defining the context of their sensitivities being global as opposed to purely local. There is also a need for better non-parametric statistics to define over embedding spaces that capture properties of frequency association and cluster structures.

---

[9]*PP* models do not generate significant frequency concentration as Figure 4 (*middle*) shows.

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

## A  EXPERIMENTAL DETAILS AND DATASETS

### A.1  EXPERIMENTAL DETAILS

This subsection presents the experimental details of figures shown in Section 2. For each of the model, we generate $n = 2000$ points in $d = 3$ dimension space. All models use random seed 17, which is implemented by numpy.random.RandomState(17). Other parameter settings of seven models are as follows.

- **Gaussian model.**  Each data point is from $\mathcal{N}(\mathbf{0}, \boldsymbol{I}_3)$.
- **Preferential Placement (PP) model.**  The probability of updating the frequency of existing nodes is set to $q = 0.85$. The parameter of exponential distribution of radius is $\beta = 1.0$.
- **Directional PP (DPP) model.**  The parameter $\kappa$ is set to $1.0$. The parameter of exponential distribution of radius is $\beta = 2.8$ and the probability of updating the frequency of existing nodes $q$ where we set $q = 0.85$.
- **BA-Gravity.**  The parameter of generating $m$ new edges is set to 20. The initial points are all from 3 dimensional Gaussian distribution.
- **BA-Centroid-NW, BA-Centroid-EW.**  We share the same parameter setting as BA-Gravity: the parameter of generating $m$ new edges is set to 20. The initial points are all from 3 dimensional Gaussian distribution.

### A.2  DATASETS

Table 2 lists word embedding datasets used in this paper, including multiple-languages from different corpus and dimensions. More specifically, GloVe-CC has two versions where one is trained from 840B tokens of English Common Crawl dataset while the other one is trained from 42B.

| Embedding | Source | Corpus | # Embedding | Dim. (d) |
|---|---|---|---|---|
| Fasttext-Wiki | (Bojanowski et al., 2017) | Wikipedia | 294 (Languages) | 300 |
| Fasttext-CC | (Grave et al., 2018) | Common Crawl, Wikipedia | 157 (Languages) | 300 |
| Fasttext-Aligned | (Joulin et al., 2018) | Wikipedia | 44 (Languages) | 300 |
| Polyglot | (Al-Rfou et al., 2013) | Wikipedia | 137 (Languages) | 64 |
| GloVe-Twitter | (Pennington et al., 2014) | Twitter | 1 (English) | 25, 50, 100, 200 |
| GloVe-CC | (Pennington et al., 2014) | Common Crawl | 2 (English) | 300 |
| GloVe-Wiki | (Pennington et al., 2014) | Wikipedia, Gigaword 5 | 1 (English) | 50, 100, 200, 300 |
| word2vec | (Mikolov et al., 2013b) | Google News | 1 (English) | 300 |
| word2vec-Poly | (Park, 2016) | Wikipedia | 29 (Languages) | 100, 200, 300 |
| BERT | (Devlin et al., 2018) | Wikipedia | 1 (English) | 786 |

Table 2: Word embedding datasets.

| | Source | Nodes / Edges | Description |
|---|---|---|---|
| Coauthor | (Guo et al., 2021) | 49,755 / 755,446 | Citation network |
| Wikipeople | (Skiena & Ward, 2014) | 214,010 / 622,255 | Wiki hyper-link network |
| Patent | (Guo et al., 2021) | 46,753 / 851,464 | Citation network |
| PPI | (Yue et al., 2020) | 3,890 / 76,584 | Biology network for protein interaction |

Table 3: Graph datasets

For each graph, we applied graph embedding algorithms ($dim = 64$) as following:

- Random walk based DeepWalk Perozzi et al. (2014),
- Random projection based FastRP Chen et al. (2019),
- First-order and second-order proximity based LINE Tang et al. (2015)
- Locally linear embedding LLE Roweis & Saul (2000).

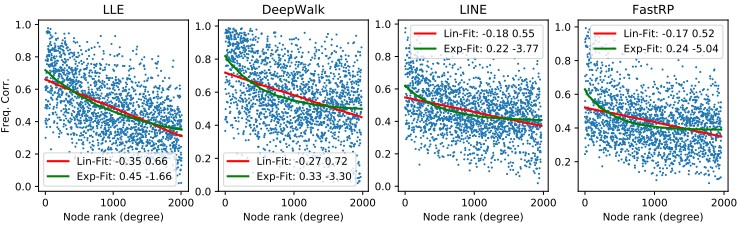

Figure 5: Frequency concentration of four distinct embeddings of the Wikipeople graph. The observed frequency concentration is similar even though trained on different graph embedding algorithms (LLE, DeepWalk, Line, FastRP).

## B MORE DETAILS IN LANGUAGE FREQUENCY CONCENTRATION

### B.1 FREQUENCY CONCENTRATION WITH RESPECT TO $k$ AND $d$

**Frequency concentration of different $k$ and $d$.** The embedding dimension $d$ and number of neighbors $k$ will affect the linear and exponential fit of frequency correlation. To see this, we study how the parameter $k$ and $d$ affect the frequency correlation on embeddings of GloVe-Twitter and GloVe-Wiki-6B. The comparison of frequency correlation w.r.t different dimensionality ($d$) and different size of neighbors ($k$). The GloVe embeddings considered in this figure are trained on Twitter datasets. $d$ is from $\{25, 50, 100, 200\}$ and $k$ is from $\{5, 10, 15\}$. We present the results in Table 4-5. When $k$ increases, the linear coefficient of Lin fit also increase.

|          | $d = 25$ | $d = 50$ | $d = 100$ | $d = 200$ |
|----------|----------|----------|-----------|-----------|
| $k = 5$  | -.533    | -.537    | -.548     | -.615     |
| $k = 10$ | -.510    | -.515    | -.526     | -.596     |
| $k = 15$ | -.496    | -.501    | -.512     | -.583     |

Table 4: Coefficients of Linear fit for GloVe Twitter Embedding.

|          | $d = 50$ | $d = 100$ | $d = 200$ | $d = 300$ |
|----------|----------|-----------|-----------|-----------|
| $k = 5$  | -.753    | -.907     | -.822     | -.802     |
| $k = 10$ | -.734    | -.890     | -.802     | -.782     |
| $k = 15$ | -.721    | -.879     | -.790     | -.770     |

Table 5: Coefficients of Linear fit for GloVe Wiki Embedding.

### B.2 FREQUENCY CORRELATION WITH RESPECT TO DIFFERENT LANGUAGES

We consider the frequency correlation learned from different languages in Figure 6.

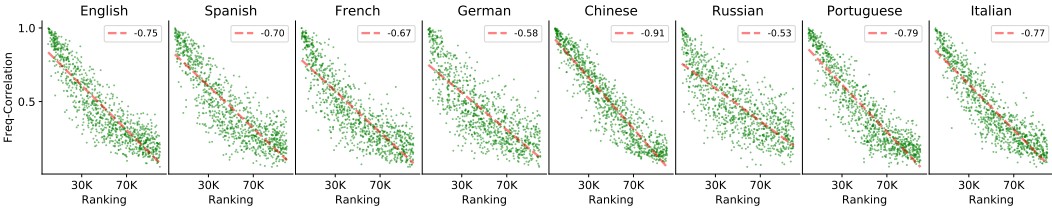

Figure 6: Frequency correlation of Polyglot embedding (Al-Rfou et al., 2013) as a function of ranking of words frequency. Each embedding dimension is $d = 64$ and $k = 10$.

### B.3 FREQUENCY CONCENTRATION FOR BERT CONTEXTUAL WORD EMBEDDING

We sample 30K unique words from the most frequent words in GLoVe (English) embeddings, and obtain their contextual embeddings from 250K sentences in Wikipedia's text [10]. Specifically, we extract the last layer of BERT output as the intermediate word embeddings. Since one word may appear multiple times in different sentences, its intermediate embedding may vary. So we average all the collected intermediate embeddings for each word as its final word representation, then feed them to our frequency concentration and cluster velocity analysis.

For example, the word "embeddings" is tokenized to be [ 'em', '##bed', '##ding', '##s' ] in BERT's tokenizer. We extract the embeddings of the first sub-token ('em') as the word-level intermediate representation in different contexts (We adopt this practice from Zhou et al. (2021) ). Then average them up as the final contextual word embeddings for the word 'embeddings'. Figure 7 shows the BERT embedding also shares consistent frequency concentration as other static embeddings do.

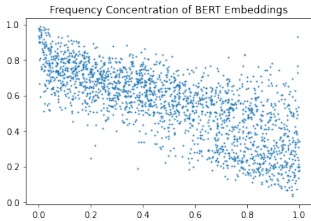

Figure 7: Frequency Concentration was also obseved in contextual word embeddings

## C PREFERENTIAL PLACEMENT (PP) MODEL

Algorithm 2 presents the generating process of PP model.

---

**Algorithm 2** Preferential Placement (PP) model

---

1: **Input:** $n$ the total frequencies of data points, $\beta$ the parameter for Exp distribution, and $q$ the probability of increasing the frequency of an old word
2: $\boldsymbol{r} \sim \text{Exp}(\beta, n)$ ▷ generate $n$ samples of radius from distribution $\text{Exp}(\beta)$
3: $\boldsymbol{\theta} \sim \text{U}(0, 2\pi, n)$ ▷ generate $n$ samples of angle from distribution $\text{U}(0, 2\pi)$
4: $\boldsymbol{W} = \{\boldsymbol{w}_0 : 1\}$ where $\boldsymbol{w}_0 = \boldsymbol{0}$ ▷ a dictionary of word-frequency pairs
5: **for** $t \in \{1, 2, \ldots, n\}$ **do**
6:   randomly select an word $\boldsymbol{w}_i$ from $\boldsymbol{W}$ with a prob. proportional to its frequency.
7:   **if** $\text{Bin}(1, q) == 1$ **then**
8:    $\boldsymbol{W}[\boldsymbol{w}_i] = \boldsymbol{W}[\boldsymbol{w}_i] + 1$ ▷ increase the frequency of an existing word $\boldsymbol{w}_i$ by 1
9:   **else**
10:    $\boldsymbol{w}_t = \boldsymbol{w}_i + [r_t \cos(\theta_t), r_t \sin(\theta_t)]^\top$ ▷ create a new word based on $\boldsymbol{w}_i$
11:    $\boldsymbol{W}[\boldsymbol{w}_t] = 1$ ▷ add this new word into word-frequency pairs.
12:   **end if**
13: **end for**
14: Return $\boldsymbol{W}$

---

## D DIRECTIONAL PREFERENTIAL PLACEMENT (DPP) MODEL

The exponential distribution is used like before. The von Mises-Fisher distribution, which generalizes the uniform distribution used earlier, for $(d-1)$-dimensional sphere is given by

$$f_{\text{vMF}}(\boldsymbol{x}|\boldsymbol{\mu}, \kappa) := \frac{\kappa^{d/2-1}}{(2\pi)^{d/2} I_{d/2-1}(\kappa)} \exp\left\{\kappa \boldsymbol{\mu}^\top \boldsymbol{x}\right\}, \tag{3}$$

---

[10]Wikipedia text is from Huggingface Wikipedia-English Dataset

where random variable $x \in \mathbb{R}^d$, $\|x\| = 1$; $\mu$, $\|\mu\| = 1$, is the mean direction parameter; $\kappa \geq 0$ is a concentration parameter; $d \geq 2$; and $I_{d/2-1}(\kappa)$ is the modified Bessel function of the first kind at order $d/2 - 1$. Algorithm 3 presents the DPP model.

---

**Algorithm 3** Directional Preferential Placement (DPP) model

---

1: **Input:** $n$ the total frequencies of data points,
$\qquad\qquad$ $\beta$ the parameter for Exp distribution,
$\qquad\qquad$ $q$ the probability of increasing the frequency of an old word
2: $X \sim f(x|\mu_i, \kappa)$ //$X$ is the direction matrix from one vMF model where $x_i$ is $i$-th row of $X$ and a specific direction randomly generated from Wood (1994)
3: $W = \{w_0 : 1\}$ where $w_0 = 1/\|w_0\|$ $\quad \triangleright$ a dictionary of word-frequency pairs
4: **for** $t \in \{1, 2, \ldots, n\}$ **do**
5: $\qquad$ randomly select an word $w_i$ from $W$ with a prob. proportional to its frequency.
6: $\qquad$ **if** $\mathrm{Bin}(1, q) == 1$ **then**
7: $\qquad\qquad$ $W[w_i] = W[w_i] + 1$ $\quad \triangleright$ increase the frequency of an existing word $w_i$ by 1
8: $\qquad$ **else**
9: $\qquad\qquad$ $w_t = w_i + r_t \cdot x_i^\top$ $\quad \triangleright$ create a new word based on $w_i$
10: $\qquad\qquad$ $W[w_t] = 1$ $\quad \triangleright$ add this new word into word-frequency pairs.
11: $\qquad$ **end if**
12: **end for**
13: Return $W$

---

# E    DETAILS IN MODEL PARAMETER SENSITIVITY

Figure 8 shows the histogram of results of each run. We repeat the experiments for 5 times with different random seed for each model configuration, where take the combination of hyper-parameters. The ranges of hyper-parameter are listed below:

```
n = 2000; d = 300; random_seed = [0,1,2,3,4]
["model-uniform"] = {}    ["model-single-gaussian"] = {}
["model-multi-gaussian"] = {"num_cluster": [2,4,8],"cluster_std":[0.01,0.1,1.0]}
# q: probability of increarsing frequency
# exp_lambda: the scale parameter to control radius
# m: the parameter of generating m new edges
["model-pref-placement"] = {   'q':[0.1,  0.3,  0.5,  0.7,  0.8,  0.9], 'exp_lambda' : [1,2,4,8,16,32,64,128]}
["model-dir-pref-placement"] = { 'q':[0.1,  0.3,  0.5,  0.7,  0.8,  0.9], 'exp_lambda':[1,2,3,4],
                 "kappa":[32,64,128],  "opt":["positive-weighted","negative-weighted"]}
["model-ba-gravity"] = {'m':[10,  20,  30,  40,  50,  60]}
["model-ba-centroid-edge-weighted"] = {'m':[10,  20,  30,  40,  50,  60]}
["model-ba-centroid-edge-average"] = {'m':[10,  20,  30,  40,  50,  60]}
["model-ba-centroid-node-average"] = {'m':[10,  20,  30,  40,  50,  60]}
["model-ba-centroid-node-weighted"] = {'m':[10,  20,  30,  40,  50,  60]}
["model-ba-embedded"] = 'm':[10],'num_walks':[3,5],  walk_len': [5,10],  'window': [5]}
```

# F    MORAN'S $I$ AND GEARY'S $C$

We start with the definitions. Each unit $y_i$ is characterized by its importance (e.g., frequency) and $d$-dimensional location. Another important ingredient is a spatial weights matrix $W$ with elements $w_{ij}$. It is either an adjacency matrix (binary, in $\{0, 1\}$), or similarity matrix (continuous, in $[0, 1]$).

Now, Moran's $I$ is defined as:

$$I := \frac{N}{\sum_i \sum_j w_{ij}} \frac{\sum_i \sum_j w_{ij}(y_i - \bar{y})(y_j - \bar{y})}{\sum_i (y_i - \bar{y})^2}.$$

Then, $I \in [-1, 1]$; $I < 0$ means negative autocorrelation (a checkerboard pattern), $I > 0$ means positive autocorrelation (same-color squares on one side).

Geary's $C$ is defined as:

$$C := \frac{N - 1}{2(\sum_i \sum_j w_{ij})} \frac{\sum_i \sum_j w_{ij}(y_i - y_j)^2}{\sum_i (y_i - \bar{y})^2}.$$

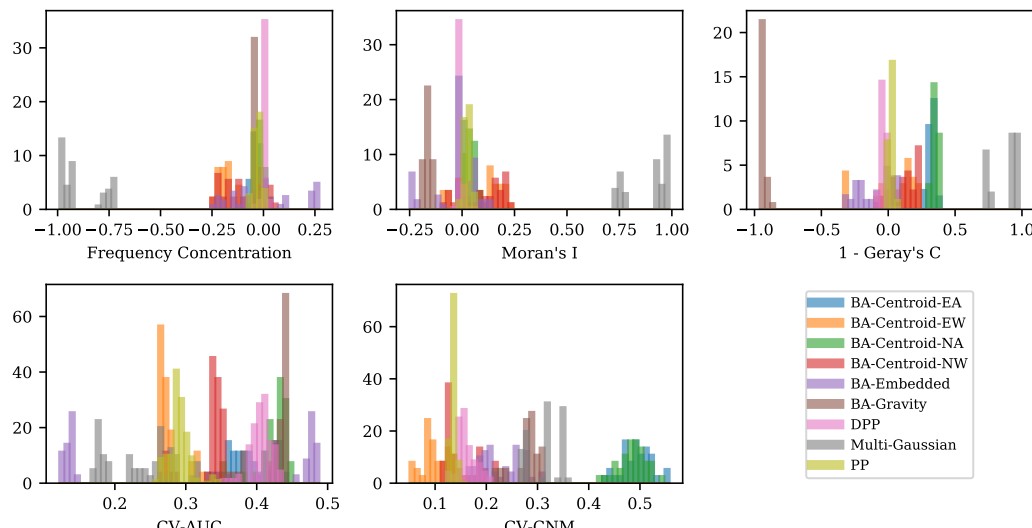

Figure 8: The PDF of discussed statistics for each model running with 5 random seeds and distinct model hyper-parameter combinations as listed above. ($n = 2000, d = 300$)

For a more intuitive interpretation, it may be easier to use $C' := 1 - C$. Then, $C' \in [-1, 1]$; $C' < 0$ means negative autocorrelation (checkerboard), $C' > 0$ means positive autocorrelation (same-color squares on one side).

We present the correlation between our proposed Frequency Concentration (FC) and Moran's $I$ as well as with Geary's $C$ in Table 6 and Figure 9.

| Spearman corr. | FC | Moran-I | 1-Geary's |
|---|---|---|---|
| FC | 1.000 | -0.810 | -0.450 |
| Moran-I | -0.810 | 1.000 | 0.662 |
| 1-Geary's | -0.450 | 0.662 | 1.000 |

Table 6: Our proposed Frequency Concentration (FC) index is well correlated with classic Moran's $I$ and Geary's $C$.

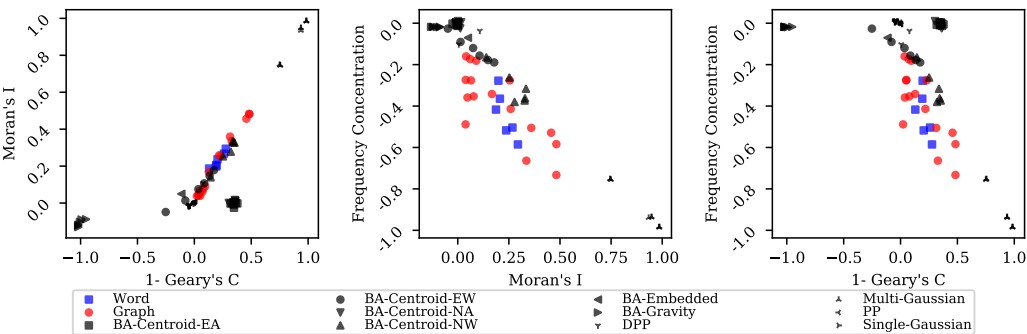

Figure 9: Moran's $I$ and Geary's $C$ are well correlated, and so does our proposed Frequency Concentration.

