# OpenReview forum: "Why do embedding spaces look as they do?"
_ICLR.cc/2022/Conference — ICLR 2022 Submitted_

### Official Review · Reviewer_bApa · 2021-10-22

**Correctness:** 2
**Technical Novelty And Significance:** 3
**Empirical Novelty And Significance:** 2
**Recommendation:** 3
**Confidence:** 4

**Main Review:**

The strengths:
S1: This paper studies an interesting problem, to analyze structure evolution from the geometry in embedding space.
S2: Two novel measures are proposed.
S3: This paper is well-written.

The weaknesses:
W1: There is no problem formulation. It's hard to know what is the computational problem studied in this paper.
W2: It's not clear how do the proposed measures answer the raised problem.
W3: There is a lack of discussion of the rationale of the proposed measures.
W4: There is a lack of empirical validation for the proposed measure.

I have some concerns about this paper. There is no problem formulation. It's hard to know what is the computational problem studied in this paper. It seems that the goal of this paper is only to reveal and explain preferential attachment. It's not clear how do the proposed measures answer the raised problem. What is the intuition of the proposed measure in terms of analyzing structure evolution?  There is a lack of discussion of the rationale of the proposed measures. There is a lack of empirical validation for the proposed measure.

**Summary Of The Paper:**

This paper proposes two measures of word/graph embedding to character the evolution of word/graph, i.e., frequency concentration and clustering velocity. Many existing graph generation models are surveyed. The proposed measures are calculated on many real-world datasets.

**Summary Of The Review:**

This paper studies an interesting problem and proposes two novel measures. This paper lacks some key elements, such as problem formulation, rationale discussion, and empirical validation. As its too many weaknesses, I have to recommend a reject.

---

> ### Author Response · Authors · 2021-11-21
> **Response to Reviewer bApa**
>
>
> Thanks for your review. Here is our response to your concerns.
>
> -----Part One-----
> -----
>
> -----
>
> **Weakness 1** : " ... no problem formulation... It seems that the goal of this paper is only to reveal and explain preferential attachment."
>
> **Response**:  The goal of our paper is indeed to document and explain the origins of the structures inherent in observed embeddings.    Our hypothesis is that there are fundamental mechanisms at work in the evolution of language and societal knowledge creation (and possibly other similar phenomena), and we use embeddings and generative models (oftentimes relying on preferential attachment mechanisms) as tools to study such processes.
>
> Since this is a new problem domain, there are no standard methods for measuring, analyzing and summarizing the embedding spaces. Thus much of our efforts involve the formulation of robust metrics to compare pairs of embeddings on different entity universes, and to propose interpretable generative processes to best fit the properties of real-world embeddings under those proposed metrics.  Thus we gain instructive clues on the fundamental processes generating the structure that is available in data to be encoded in embedding spaces.
>
> -----
>
> **Weakness 2**: ...how do the proposed measures answer the raised problem.
>
> **Response**: Our proposed measures are for evaluating the structural similarity between the real-world embedding space and the space from interpretable generative processes -- which requires robust metrics on unlabeled embedding spaces (i.e., there is not a 1-1 correspondence between the two). If a given generative process could produce very similar measures when compared to the real-world embeddings, we postulate that this specific generative process is able to explain why embedding spaces look as they do.
>
> -----
>
> **Weakness 3**: ".., lack of discussion of the rationale of the proposed measures. What is the intuition of the proposed measure in terms of analyzing structure evolution? "
>
> **Response**:  Embeddings have value for machine learning because they make explicit cluster structures among the elements in the space.  Such structures are inherent in all domains (e.g. nouns and verbs and colors in language vocabularies, and domains such as science and entertainment and history in Wikipedia).  We are interested in why language and cultural artifacts have the clustering structures they do, and why.
>
> We are exploring new subject domains such as linguistics and culture with the tools inspired by representation learning and other fields such as geographic analysis. Existing embedding tools seem to be inappropriate for analyzing novel data patterns that we observe in the data. In other words, we have to simultaneously formulate tools for our analysis and to specify data regularities these tools should be capturing.  At this exploratory stage we indeed rely on trial and error for discovering measures that seem to be informative about the patterns in the real-world data.
>
> As detailed in Section 3 and Section 4, we measure the spatial correlation and cluster structure implicitly preserved in the embedding space.   We believe that the proposed measures capture the common properties in the real-world embeddings in an unlabeled manner (comparing two structures does not require a common set of entities, but can be completely disjoint). Thus we can compare observed embeddings to model generated spaces, as shown in Figure 3 and Figure 4.  This provides an empirical validation of the hypothesis that language generation processes reflect phenomena like preferential attachment and placement.
>
> The intuition that we can offer for frequency concentration measure is that it is a version of spatial correlation coefficient with nearest-neighbor-based spatial distances. The intuition for clustering velocity is that the connectivity of graphs induced by embeddings that it measures is yet another alternative way to characterize spatial distances: the velocity is **not** meant to measure structure evolution per se, but instead it exploits the fact that the merging process will happen faster for a set of points will less clusters.

---

> > ### Author Response · Authors · 2021-11-21
> > **Response to Reviewer bApa**
> >
> > -----Part Two-----
> > -----
> >
> >
> > -----
> >
> > **Weakness 4**: "... lack of empirical validation for the proposed measure."
> >
> > **Response**:
> > - For the measures we borrow from other fields (such as spatial autocorrelations), empirical validity is a long-settled issue.
> >
> > - The validity of our newly proposed measures is verified on model-generated data: they are able to differentiate between datasets generated by models with different characteristics. (For the real-world data, we do not know the ground truth, hence there is no criterion to tell if our proposed measures work as required.) We agree that at this point we are providing first quantitative conclusions, issues such as statistical significance and performance in broad Monte Carlo simulations is beyond the scope of this work (which focuses on setting the stage for making progress on tackling the questions we asked at the outset) .
> >
> > - In Appendix {B, E}, we also try different model and measure parameters, and observed consistent patterns empirically. This is a strong evidence to prove our results are not cherry-picked anecdotes from invalid measures .
> >
> > -----
> >
> > Again, thanks for the reviews. Hope our responses answer your questions

---

### Official Review · Reviewer_XDaN · 2021-10-27

**Correctness:** 4
**Technical Novelty And Significance:** 3
**Empirical Novelty And Significance:** 2
**Recommendation:** 5
**Confidence:** 2

**Main Review:**

Strengths

1. This paper studies data generation processes with the help of embedding spaces. It is an interesting problem, and I think the idea of bridging embedding spaces and data generation processes is worth further exploration.
2. The authors give a detailed illustration of several data generation methods.



Weakness
1. The title is a bit confusing. The authors ask a question "why do embedding spaces look as they do" in the title. However, it seems that they did not answer it throughout this submission.
2. The submission is not self-contained. For example, at the bottom of Page 6, the authors put an important figure in the appendix. However, as noted by the ICLR committee, reviewers are not required to read appendices.
3. The linear correlations in the experiments seem to be weak, especially in Figure 2. The authors may want to calculate the Pearson's correlation between Freq. Concentration and ranking.
4. As described in the conclusion section, experiments in this work are not sufficient to identify a single dominant model for reasoning about the origins and structures of natural embedding spaces. I agree that this work gives a good start for an interesting problem. However, since the authors did not give a clear conclusion, the technical contribution of this submission seems to be weak.


**Summary Of The Paper:**

This paper tries to explore the natural processes that generate new knowledge or concepts. Specifically, the authors propose two metrics to characterize embeddings trained on different datasets. Then, they compare some synthetic data generated by models and real-world data according to the aforementioned metrics. Finally, they conclude that the real-world data can be well simulated by a certain generative model.

**Summary Of The Review:**

This paper studies an important and interesting problem. However, it does not provide a good enough solution to the problem to meet the bar of ICLR.

---

> ### Author Response · Authors · 2021-11-21
> **Response to Reviewer XDaN**
>
> We would like to thank the reviewer for the comments and suggestions. Here is our response to your concerns and suggestions.
>
> -----
>
> **Weakness1**: "The title is a bit confusing. The authors ask a question "why do embedding spaces look as they do" in the title. However, it seems that they did not answer it throughout this submission."
>
> **Response**: We don’t completely answer this question, but we shed considerable light on this question!   We show that some generation models exhibit properties apparent in real embeddings while others do not.
>
> Why embedding spaces look as they do is the focus of our study -- what are the generative processes behind language formation and knowledge generation that produce the structure to be encapsulated in embeddings.
>
> -----
>
> **Weakness2**: " The submission is not self-contained. For example, at the bottom of Page 6, the authors put an important figure in the appendix. However, as noted by the ICLR committee, reviewers are not required to read appendices."
>
> **Response**: We are sorry for not including the details due to the space limitations. Actually Table 1 (in the main body) includes the information in Figure 6 (which we put it in the appendix). We've updated the draft, emphasized it and rearrange the words.
>
> -----
>
> **Weakness3**: "The linear correlations in the experiments seem to be weak, especially in Figure 2. The authors may want to calculate the Pearson's correlation between Freq. Concentration and ranking."
>
> **Response**: Thanks for your suggestion. Strength of Pearson’s correlation very much depends on the stochastic aspects of the phenomena in question (e.g., on the relative amount of noise), and we believe this is an interesting, but second-order issue. A question of the first order is a qualitative pattern in their relationship, i.e. a slope (abstracting away from the fact that linearity is only an approximation in this case).
>
> Pearson's correlation measures the strength of  linearity, but may not reflect how ‘steep’ the linear relation is. For example, the points lying at y = -0.5x and y=-10x will both produce 1.0 Pearson’s correlation (for variable X and Y), but the latter one is steeper, which we want to measure.
>
> -----
>
> **Weakness3**: "As described in the conclusion section, experiments in this work are not sufficient to identify a single dominant model .... , since the authors did not give a clear conclusion, the technical contribution of this submission seems to be weak."
>
> **Response**: Our paper makes a good start at an interesting problem.   It is designed to provoke fairly deep questions about how language formation and knowledge generation processes work.    Our paper will be successful if it starts new lines of work and thought.
>
> Since these questions are new and difficult, our results are not definitive, but they are an important start.   Rather than giving a definite final answer, we focus on problem formulation, documenting some first empirical regularities and proposing several potentially useful and revealing theoretical models.    We find strong clues that two of our proposed models (BA-Centroid-{NW,EW}, DPP) fit the observed data, and we consider those results with observed patterns as our technical contributions.
> Hopefully this will lead to substantial future work around this open problem, and we hope this paper will serve as a pioneer to encourage more future discussions.
>
> -----
>
> Again, we would like to thank you for the reviews. Hope our responses answer your questions.

---

### Official Review · Reviewer_rVxQ · 2021-11-03

**Correctness:** 3
**Technical Novelty And Significance:** 3
**Empirical Novelty And Significance:** 3
**Recommendation:** 5
**Confidence:** 3

**Main Review:**


Strengths.

In general, I find the question the authors try to address is interesting. The author's methodology of approaching this problem seems fine.

Weaknesses.

- One concern I have is the embedding the authors use. Now NLP community is adapting to embeddings from pre-trained models, e.g., BERT. It seems to me that analyzing these contextual embeddings would be better to understand the question the authors propose.

- Somehow I feel like the evaluation between the embeddings from the generative model and from the real-world is rather weak. I guess another more straightforward way is to show the performance on downstream tasks using the generated embeddings.

**Summary Of The Paper:**


In this paper,  the authors try to study if we can learn about the human's process of generating new ideas or concepts from embeddings. The authors first define a set of models for generating embeddings. Then the authors observe two properties: (1) frequency concentration, and (2) cluster velocity. The authors finally compare the embeddings from generative models with the embeddings from real-world data driven methods.

**Summary Of The Review:**


In sum, I think the authors look at an important problem. But there are some limitations of this study, e.g., not looking at contextual embeddings.

---

> ### Author Response · Authors · 2021-11-21
> **Response to Reviewer rVxQ**
>
> We would like to thank the reviewer for the comments and suggestions. Here is our response to your concerns.
>
> -----
>
> **Weakness 1**: "One concern I have is the embedding the authors use... , e.g., BERT. It seems to me that analyzing these contextual embeddings would be better ...."
>
> **Response**: Thanks for your suggestion. We conducted an experiment on BERT embeddings, and our new results have been updated in figure {3,4,7}, also Appendix B.3 in the updated draft.   They show that the observed phenomenon in BERT embeddings is consistent with what we see in other embeddings.  This is in line with our belief that successful representation learning methods capture real structure inherent in linguistic or cultural domains -- making them a unique way to study linguistic and cultural generation processes.
>
> -----
>
> **Weakness 2**: "Somehow I feel like the evaluation between the embeddings from the generative model and from the real-world is rather weak. I guess another more straightforward way is to show the performance on downstream tasks using the generated embeddings."
>
> **Response**: We agree that the usability and explanation are both important. But we have discovered important gaps in scientific understanding of the patterns in information and knowledge produced by the human civilization (e.g., language, Wikipedia). Thus, the goal of this paper is to understand and investigate what kind of (simple and intuitive) theoretical generative process can create embedding spaces that exhibit commonly reoccurring  properties observed in the real-world embeddings. Improved embedding models stemming from our findings will hopefully be formulated in the future work. (*We also answered the similar question raised by Reviewer  6Hzd in Weakness 2.1.*)
> To our best knowledge, a similar research paradigm could be found here:
> - Leskovec, J., Kleinberg, J., & Faloutsos, C. (2007). Graph evolution: Densification and shrinking diameters. ACM transactions on Knowledge Discovery from Data (TKDD), 1(1), 2-es.
>
> -----
>
> Again, we would like to thank you for the reviews. Hope our responses answer your questions.

---

### Official Review · Reviewer_6Hzd · 2021-11-04

**Correctness:** 3
**Technical Novelty And Significance:** 2
**Empirical Novelty And Significance:** 2
**Recommendation:** 5
**Confidence:** 4

**Main Review:**

(Strengths)
This paper proposes interesting approaches to review popular static word embeddings. The authors try to reconstruct a number of different embeddings based on different node generation models, edge generating models, and their combinations.

1) It is exciting to read the authors’ endeavor on revisiting the old successful embeddings that brought a wide range of breakthrough in NLP.

2) Their approach of viewing embeddings purely as a result of node-only generations vs a product of graph that has both node and edge generations continued by graph embeddings
 seems to be thorough instrument to investigate the embedding spaces.

3) Their findings of frequency concentration (not only highly frequency terms getting richer) and clustering velocity (but also they tend to be together) is useful to support an important observation: there are significant amount of frequency information in learned embeddings.

4) The authors revisit many interesting concepts and ideas in different fields like geography that covers spatial relationships and corresponding philosophy.


(Weaknesses)
1) The primary concern is that reading this paper may not add useful intuitions in the era of contextual embeddings and large-scale language models. If the paper suggests any type of treatment that may alleviate too much dependency on the frequency information; or that can improve the performance of state-of-the-art language models by feeding better initial embeddings, the research will become a gold. But the current draft itself is less likely to impact on our community.

2) All the models and experiments are designed and operated rather passively by tuning the hyperparameters and parameters then comparing the results with the popular learned embeddings. It is of course useful to have such models analyses. However it is more natural for Machine Learning community to fit each model into the training portion of data and validate it on the hold-out set before making comparisons. Otherwise, we cannot say whether each suggested model actually has some degree of generalizability of the phenomena of interests.

3) Indeed, we already understood in both high-level and low-level that these static embeddings largely encode frequency information though they were often argued to encapsulate semantic information. Usual definition of the context in computational linguistics is a set of words in the sliding window of each word, which inevitably connects to learning frequential/distributional similarities rather than precisely semantic and syntactic similarities. It was also studied that frequent words are more frequently updated in SGD framework, sharing the large norms. If we use Gaussian like kernel, it will be not surprising to see these are grouped.




**Summary Of The Paper:**

This paper tries to demystify the power of embedding and embedding space, trying to connect the structure of learned embeddings and knowledge generation process. The authors tested Gaussian model, Preferential Placement (PP) model, and Directional Preferential Placement (DPP) model for node generations. The authors also tried to learn embeddings by graph representation learning algorithms after running BA models with several variances with gravitational movement and centroid node/edge weighting.


(Main Contributions)
1) Propose reasonable processes that can explain the evolution of knowledge embeddings in terms of preferential attachment and attractive-repulsive force.

2) Compare observed and generated embedding spaces via non-parametric statistics, identifying frequency concentration and clustering velocity properties.

3) Evaluate and find the best generative process: incremental insertion process with spatial context-dependent preferential attachment.


**Summary Of The Review:**

(Questions)
1) Why do you believe that embeddings are optimized to preserve the natural clusters of the real-world knowledge in its metric space? At least the three embeddings used in the paper never explicitly optimize any clustering objectives. They want the words closely in the raw texts locate closely in the Euclidean space.

2) Geometry often means the characteristics of the manifold or topological essence defined by a collection of open sets (that will indeed define suitable metric for metric spaces). Some expressions about geometry or invariance would be too subjectively used.

3) It is interesting to see how clustering velocity is defined as area under the curve in the graph of increasing number of connected components. Different types of graphs (maybe different scale-freeness and degree distributions) naturally differs in this clustering velocity, but it is a bit doubtful that this metric can be drastically different even for two graphs with similar degree distributions by adversarially tweaking couple of local connections. As both models and new metrics are proposed, there must be a natural question about the stability and sensitivity of this metric.

4) More active form of contributions that can impact modern representation learning and transfer-based language modeling will benefit the audiences in the community.


(Minor Comments)
A2) left figure  attach the figure reference as the actual figure is located in the previous page.

A3) figure 5 not in the main draft.

---

> ### Author Response · Authors · 2021-11-21
> **Response to Reviewer 6Hzd**
>
> We would like to thank the reviewer for the comments and suggestions. Here is our response to your main concerns and questions.
>
> ----Part One ----
> -----
>
> **Weakness 1.1**: “The primary concern is that reading this paper may not add useful intuitions in the era of contextual embeddings ...“
>
> **Response**: We seek to use embeddings as a proxy for the conceptual positioning of entities in the social and cultural world, to better understand the forces that create them and act on them.   It is an observed fact that many entities straddle multiple worlds and senses.   Although contextual embeddings are successful in decoupling the appropriate sense in a given context, we are more interested in the phenomenon of where/why we have multi-sense entities in the first place.   Thus static embeddings are the appropriate topic of study here.
>
> But as you suggested, contextual embeddings play a major role in NLP today, so we added the analysis experiment for BERT embeddings, and found the consistent patterns (updated Figure 3 and Appendix B.3). It further supports our notation that embeddings are not mere technical artifacts, but that the structure of embeddings captures invariant properties about language that hold regardless of the embedding algorithm.
>
> -----
>
> **Weakness 1.2**:“If the paper suggests any type of treatment .. or that can improve the performance ...”
>
> **Response**:Thanks for the suggestion. We believe that the incremental insertion models we propose have some utility for constructing fast embedding algorithms, but to focus on that is to miss the primary point of our paper.
> We see embeddings as an intellectual tool for studying language and knowledge creation processes, not merely an engineering tool for improving the performance of machine learning systems.   This aspect of representation learning is the novel and important issue we study in this paper.  Implications of this for better embedding models are left for the future work.
>
> -----
>
> **Weakness 2.1**: All the models and experiments are designed and operated rather passively ..... However it is more natural ... to fit each model into the training portion of data and validate it on the hold-out set before making comparisons.“
>
> **Response**:We admit that there is no prediction task in our paper, because primary interest is understanding the conceptual/historical processes generating the information which is then available for models to learn. We believe that it is the first step to understand the current embedding space before making the model performance better.
> To our best knowledge, a similar research paradigm could be found here:
> -  Leskovec, J., Kleinberg, J., & Faloutsos, C. (2007). Graph evolution: Densification and shrinking diameters. ACM transactions on Knowledge Discovery from Data (TKDD), 1(1), 2-es.
>
> -----
>
> **Weakness 2.2**:“Otherwise, we cannot say whether each suggested model actually has some degree of generalizability of the phenomena of interests.”
>
> **Response**: The common patterns observed in the word embeddings and graph embeddings (from several different real-world datasets and several different embedding algorithms) provide strong evidence to validate our hypothesis.  We also try different model parameters (in Appendix E) to show the models we present in the body of the paper are not cherry-picked anecdotes.
>
> -----
>
> **Weakness 3.1**: "Indeed, we already understood in both high-level and low-level that these static embeddings largely encode frequency information .... It was also studied that frequent words are more frequently updated in SGD framework, sharing the large norms. If we use Gaussian like kernel, it will be not surprising to see these are grouped.”
>
> **Response**: We agree that the existing works interpret the large norms as the consequence of updating more frequently in SGD framework. But here we found that the high frequency entities tend to have high concentration (high frequency entities stay closer) in the embedding space, so it’s more about spatial location (direction) rather than magnitude.
> Moreover, the graph embeddings we study include LLE (SVD) and FastRP (Random Projection): neither of these methods is based on SGD, but we observe the consistent pattern.   The message of our paper is that these patterns exist primarily because that is how language and network entities are structured by human knowledge generation processes.   That they are captured by embeddings is not a bug but a feature -- and this is an important take home message of our paper.
>
> -----
>
> **Weakness 3.2**: “If we use Gaussian like kernel, it will be not surprising to see these are grouped”
>
> **Response**: We are not sure whether we understand the above statement. However, the Gaussian-like kernel may need to have strong assumptions while ours do not make any assumptions on models.

---

> > ### Author Response · Authors · 2021-11-21
> > **Response to Reviewer 6Hzd**
> >
> >
> > ---- Part Two ----
> > ----
> >
> > ----
> >
> > For the questions:
> >
> > **Question 1**: "Why do you believe that embeddings are optimized to preserve the natural clusters of the real-world knowledge in its metric space? At least the three embeddings used in the paper never explicitly optimize any clustering objectives. They want the words closely in the raw texts locate closely in the Euclidean space. "
> >
> > **Response**: Any effective representation learning scheme must capture natural clusters that exist in its metric space, or it wouldn’t be an effective representation learning scheme in the first place.   This is independent of whether it was explicitly trained to capture the clusters in the first place.
> >
> > We have observed that many successful applications (prediction tasks) use embeddings to draw good decision boundaries with only a few layers of MLPs.  This implies that the embedding space from the representation learning process must capture the natural clusters of real-world knowledge, so that even simple decoders (e.g.: logistic regression, MLPs) find reasonable decision boundaries for the downstream classification tasks.
> >
> > -----
> >
> > **Question 2**: Geometry often means the characteristics of the manifold or topological essence defined by a collection of open sets (that will indeed define suitable metric for metric spaces). Some expressions about geometry or invariance would be too subjectively used.
> >
> > **Response**: Thanks for your suggestion. By “geometry” we mean spatial characteristics of the phenomena studied. We changed the word choices to avoid confusion.
> >
> > -----
> >
> > **Question 3**: "It is interesting to see how clustering velocity is defined as area under the curve in the graph of increasing number of connected components. Different types of graphs (maybe different scale-freeness and degree distributions) naturally differ in this clustering velocity, but it is a bit doubtful that this metric can be drastically different even for two graphs with similar degree distributions by adversarially tweaking couple of local connections. As both models and new metrics are proposed, there must be a natural question about the stability and sensitivity of this metric."
> >
> > **Response**: Thanks for bringing up this important question. The clustering velocity **can** differ substantially even for two graphs with similar degree distribution.   Consider the following two graphs where each of n vertices have degree of sqrt(n):
> >
> >  - An Erdos-Renyi graph with n sqrt n edges.   The large component will subsume most vertices after a linear number of insertions
> >
> >  - Sqrt(n) cliques, joined in a path with single edges.   We still expect sqrt(n) components after a linear number of insertions.
> >
> > We agree that a suitable and robust metric is the key to our research. Without a valid metric, we cannot measure the characteristic of real-world embeddings, or claim that the proposed generative process best explains (or at least mimics) the real-world embeddings.
> >
> > From our empirical results, we found that all real-world embeddings have similar metric values (Freq. concentration and Cluster velocities), and several known less possible models (Gaussian, Uniform models) do not fall in the same range. That makes us believe the proposed metrics are valid, and sensitive to different spaces, though they’re all empirical observations.
> >
> > -----
> >
> > **Question 4**:  More active form of contributions that can impact modern representation learning and transfer-based language modeling will benefit the audiences in the community.
> >
> > **Response**: As stated above we believe that our local incremental models do have something to contribute to creating very fast embedding algorithms(e.g,: for embedding post-processing), but this is not the focus of the submitted paper. We believe ICLR is an appropriate venue for our paper on embeddings as a unique tool to understand linguistic and historical processes, and demonstrate some first findings in that area, not just engineering improvements
> >
> >
> > -----
> >
> > Again, we would like to thank you for the reviews. Hope our responses answer your questions.

---

### Decision · Program_Chairs · 2022-01-20

**Decision:**

Reject

**Comment:**

The authors of this work introduced new metrics for node embedding that can measure the evolution of the embeddings, and compare them with existing graph embedding approaches, and experimented on real datasets.

All reviewers agreed that the work addresses interesting problem and that the proposed measures are nove, but there are too many flaws in the initial version of the paper, and despite the thorough responses of the authors, it is believed that there are still too many open questions for this paper to be accepted this year ICLR.